# Characterization of white and red sorghum flour and their potential use for production of extrudate crisps

Julia Guazzelli Pezzali[¤⊚], Anu Suprabha-Raj[‡], Kaliramesh Siliveru[‡], Charles Gregory Aldrich[*⊚]

Department of Grain Science and Industry, Kansas State University, Manhattan, Kansas, United States of America

⊚ These authors contributed equally to this work.
¤ Current address: Department of Animal Biosciences, University of Guelph, Guelph, Ontario, Canada
‡ These authors also contributed equally to this work.
* aldrich4@ksu.edu

**Data Availability Statement:** All relevant data are within the paper and its Supporting Information files.

## Abstract

In the human food industry, the wheat-free market sales have increased over the years due to awareness of wheat gluten allergy and celiac disease. Sorghum is a gluten-free grain with great potential to address shortcomings in this market. The aim of this study was to evaluate the milling process and flour quality of one white and one red sorghum varieties and evaluate extrusion as a potential process to produce sorghum crisps. The white and red sorghum grains were milled into flour in three production cycles. Flour quality was evaluated by determination of nutritional composition, pasting, and thermal profile. Extrusion processing of white and red sorghum flour was performed, and macrostructure of final product was evaluated. The white and red sorghum used in this study yielded similar flour content (P > 0.05). Chemical analyses revealed a higher protein and lower starch content for white sorghum than red sorghum flour (P < 0.05); however, their pasting properties did not differ. Initial and peak gelatinization temperatures were higher (P < 0.05) for red sorghum compared to white sorghum flour. Regarding particle size, white sorghum flour presented lower d10 and d50 compared to the red sorghum flour (P < 0.05). However, these differences did not impact the extrusion conditions, and white and red sorghum crisps had similar macrostructure characteristics. In conclusion, although differences in nutritional, thermal, and particle size properties were observed between the sorghum flours used in this study, changes in extrusion parameters were not needed in order to produce sorghum crisps with similar characteristics.

## Introduction

Growth in the human and pet food industry is driven by the addition of new products into the market. This often necessitates the use of new ingredients, and the development of new food forms, and processes. More than ever consumers are demanding foods that provide optimal health benefits for themselves and for their pets. For example, the wheat-free food market has edged-up due to consumers concern regarding celiac disease, gluten sensitivity, and wheat allergy

**Funding:** This research was funded by the Kansas Department of Agriculture through the U. S. Department of Agriculture Federal State Marketing Improvement Program, grant number PP35176. CGA received the award/funding. The funder had no role in study design, data collection and analysis, decision to publish, or preparation of the manuscript. https://www.ams.usda.gov/services/grants/fsmip.

**Competing interests:** The authors have declared that no competing interests exist.

[1]. Although these are diagnosed in a small percentage of the human [2] and dog populations [3], gluten-free foods now account for an important part of the market. This represents an opportunity for evaluation of novel wheat-free, gluten-free, and alternative ingredient recipes.

Sorghum is the fifth most important cereal crop grown in the world and the third most important in the United States. It has a great potential for the gluten-free market and as a healthy alternative ingredient. Some sorghum varieties are rich in phytochemicals such as phenolic acids and condensed tannins which are known to have antioxidant and antiradical activities [4]. Sorghum can be milled into flour which can be used as a major ingredient for many food applications. The use of sorghum flour has been evaluated in different systems such as cookies [5], breads [6], noodles [1], and tortillas [7]. A recent study reported that extruded sorghum flour reduced adipogenic genes, chronic inflammation, and weight gain in obese rats [8]. Development of extruded sorghum flour crisps may create a new market for this grain in the human and pet food industry. However, limited information is available on extrusion of sorghum flour. Establishing processing conditions necessary to create a consistent extrudate is essential to introduce the product into the market. Moreover, characterization of the milling process and flour quality will aid in understanding the functionalities of the raw material and provide meaningful information to the industry regarding process optimization. It is also important to assure food safety of the final product as possible Salmonella contamination may lead to recall of the product. Thus, the objective of this study was to characterize the milling process and flour quality of white and red sorghm, to determine the extrusion parameters necessary to create consistent white and red sorghum crisp, and to evaluate the presence of Salmonella in the process and final products.

## Materials and methods

### Grain processing and milling

White and red sorghum (2017 crop year) were sourced from a local farmer (Kearny County, KS, U.S.A) and milled at the Hal Ross flour mill (HRFM, Kansas State University, Manhattan, KS, U. S.A). Milling of both white and red sorghum was performed in three days totaling three replicates per treatment. The grains were cleaned and tempered to 16.5 ± 1.5% (w.b) for 19 h prior to milling. Tempering conditions were established from previous production at HRFM. The flour milling process consisted of 5 break (BK) passages, 2 sizing passages, 6 reduction passages, one quality and one tailing passage, and four purification passages. The first BK was set to net 25% release while the second and third break were set to net 75% release. Reduction roll settings were adjusted according to previous experiments [9]. Flour yield was calculated according to equation:

$$\text{Flour yield (\%)} = \frac{\text{(total flour weight, kg)}}{\text{(total sorghum to the mill, kg)}} * 100 \tag{1}$$

Sample collection was begun after 30 min of milling at a time point deemed to have reached process stability. White and red sorghum grain samples were collected from the storage silo, and the tempering bin on each milling day. The moisture content of unground grain was determined using the drying oven method: 10 g in a drying oven at 130°C for 18 h (ASABE S352.2, 1997).

### Flour characterization

Upon process stabilization within each replicate production cycle, flour samples were collected every 30 minutes totaling three subsamples per day (replicate) for each sorghum variety. Subsamples within day were composited and prepared for analysis as described below.

Flour samples were evaluated for moisture (AOAC 930.15), crude protein (AOAC 990.03), crude fat (AOAC 2003.05), ash (AOAC 942.05), total dietary fiber (AOAC 991.43, mod), and minerals including calcium, phosphorus, potassium, magnesium, sodium, sulfur, copper, iron, manganese, and zinc (AOAC 985.01; modified) at a commercial laboratory (Midwest Laboratories, Omaha, NE). Total starch and damaged starch were analyzed with commercial kits (Megazyme International Ltd, Wicklow, Ireland).

Flour pasting properties were determined using a Rapid Visco Analyser (RVA, Perten Instruments AB, Hargersten, Sweden) following AACCI Method 76–21.01. Wherein, the flour sample (3.5 ±0.1 g) was placed in an aluminum canister with distilled water (25 ± 0.1 ml) and moisture was adjusted up to 14%. The paddle was placed into the cannister, and the assembly was inserted into the RVA. The pasting temperature (PT), peak viscosity (PV), trough viscosity (TV), breakdown viscosity (BV), final viscosity (FV), set-back viscosity (SBV), and peak time were analyzed with the aid of a software (Thermocline for Windows software).

Thermal transition temperatures for gelatinization of white and red sorghum flours were assessed using a differential scanning calorimeter (DSC; Q100, TA Instruments, New Castle). Flour samples were weighed (7 ± 2 mg) in an aluminum pan with addition of distilled water (2:1, water/flour, wt/wt). Samples were heated from 10°C to 140°C at a rate of 10°C/min. The onset (To), peak (Tp), and conclusion (Tc) temperatures, and the enthalpy of gelatinization ($\Delta$H) were determined.

Particle size analysis of the red and white sorghum flours were performed by Malvern Morphology G3 (Malvern Panalytical, UK) in which 5 mm$^3$ of the flour was dispersed uniformly on the glass slide using the sample dispersion unit. Morphological parameters of all the individual particles on the predefined area was determined by the equipment. Size distribution of the particles were characterized by diameters d10, d50 and d90 which represents 10%, 50% and 90% of the particles with diameters lower than the specified value. Shape factors for elongation, aspect ratio, circularity, and convexity were also analyzed by the instrument. A detailed description of the mentioned shape parameters is given by Saad et al. [10].

## Extrusion process

Red sorghum flour (RSF) and white sorghum flour (WSF) were processed over three days using a single screw extruder with diameter of 133.35 mm and L/D ratio of 13.1:1 (model E525, Extru-Tech, Inc., Sabetha, KS, USA). Each trial day was considered a replicate totaling three replicates per treatment. Preconditioner shaft speed was set at 145 RPM with paddle configuration as follow: 1 beater in 45° forward, 49 beaters in neutral, 12 beaters in 45° reverse, 8 beaters in neutral, and 1 wiper. Extruder screw profile (Fig 1) was chosen based on preliminary trials performed on the same extruder. Extruder screw speed and knife speed were set a 425 and 1,500 RPM, respectively. No steam was added in the extruder barrel. Extruder sections (jacketed) were heated with the exception of the last (head) which was cooled to control product expansion. One die-plate with twelve-cylinder openings (1.1mm x 4.7 mm) was used. A temperature probe was inserted at the point at which extrudate exits to measure die temperature.

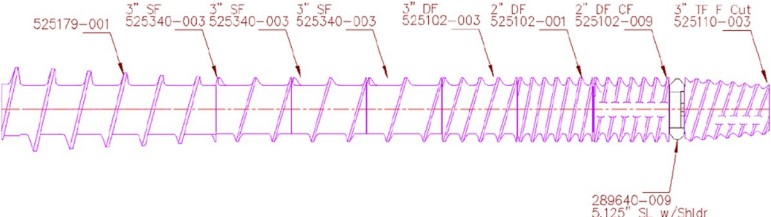

**Fig 1. Extruder screw profile used to produce expanded sorghum crisps.**

The extrudates were dried in a dual pass dryer for 7 min at 87˚C. The operator was allowed to modify processing parameters in order to achieve similar bulk density out of the extruder (~ 100g/L) for both treatments. Upon process stabilization, processing parameters and product samples out of the extruder and dryer were collected every 15 minutes. The total mass flow out of the extruder was calculated by summing the feed rate (kg/h), steam injection (kg/h), and water injection (kg/h) into the system. Product bulk density was measured using a one-liter cup. Specific mechanical energy (SME) was calculated as follow (Eq 2):

$$\text{SME (kJ/kg)} = [\tau - \tau o/100 * (N/Nr) * Pr]/m \tag{2}$$

Where $\tau$ is the % torque, or motor load, $\tau o$ is the no-load torque (34%), N is the screw speed in rpm, Nr is the rated screw speed (508 rpm), Pr is the rated motor power (37.3 kW), and m is the total mass flow in kg/s. In-barrel moisture (IBM) was calculated as described in Eq 3:

$$\text{IBM (\%)} = \frac{(mf * Xf + mps + mpw + mes + mew)}{(mf + mps + mpw + mes + mew)} \tag{3}$$

Where mf is the dry feed rate, Xf is moisture content of the feed material, mps is the steam injection rate in the preconditioner (kg/h), mpw is water injection rate in the preconditioner (kg/h), mes is the steam injection rate in the extruder (kg/h), and mew is the rate of water injected in the extruder (kg/h).

## Extrudate characteristics

Sixty extrudates from each replicate were randomly selected to assess macrostructure characteristics. Length and width were measured using a digital caliper, and weight of the same extrudates was recorded using an analytical scale (EX324N; OHAUS Corporation, Parsippany, NJ, U.S.A). Diameter was defined as the distance between the two parallel planes restricting the extrudate perpendicular to that direction, and thus length of the extrudate was considered the diameter for calculating sectional expansion index (SEI) according to the formula bellow:

$$SEI = (de * de)/(dd * dd) \tag{4}$$

Where de is extrudate diameter, and dd is die diameter.

Extrudate hardness and crispness were determined using a Texture Analyzer (Model TA-XT2; Texture Technologies Corporation, Hamilton, MA, U.S.A.). Twenty-one extrudates from each replicate were randomly selected and individually assessed for its mechanical property. Extrudates were kept in a drying oven overnight at 40˚C to equilibrate moisture content. A compression test was performed using a 25 mm cylindrical probe at a pre-test speed of 2 mm/s, test speed of 2 mm/s, a post-test speed of 10 mm/s, and strain level of 90%. The first peak fracture force, and the number of positive peaks were taken as a measure of hardness and crispness, respectively.

## Microbiological testing

Five sites at the HRFM were tested for Salmonella before and after production. Samples were collected with a sponge-stick pre-soaked in 10 mL buffered peptone water (BPW; 3M, St Paul, MN) in an area of 5X5 inches. Samples were brought to the laboratory within one hour from collection. Each sample was enriched with 50 mL of BPW (1:6 sample/ BPW, wt/wt), and incubated at 37˚C for 24 hours. Flour and extrudate samples from each sorghum variety were also tested for Salmonella in each replicate. A portion of sample (25g) was mixed with 225 ml of BPW and incubated for 20–24 h at 37˚C. After the incubation, environmental, flour and

extrudate samples were proceeded for Salmonella isolation and identification according to the standard culture method from Bacteriological Analytical Manual (BAM).

## Statistical design and analysis

The study was designed as a completely randomized block designed with day of production as a blocking factor. A total of three replicates per treatment was achieved. Analysis of variance was conducted using the GLIMMIX procedure in statistical software (SAS 9.4 Inst. Inc., Cary, NC). Sorghum variety was used as a fixed effect while day was considered a random effect. Means were separated using Fisher's LSD, and a probability of $P < 0.05$ was accepted as significant.

## Results

### Milling process

Tempering time was sufficient to bring the grain moisture from 13.74 ± 0.52 to 15.90 ± 0.99% (wb, mean ± SD), and from 14.91 + 0.56 to 15.21 + 0.73% (wb, mean ± SD) for WS and RS, respectively. The flour yield was similar ($P > 0.05$) for WS and RS (59.03 ± 8.7 and 57.02 ± 2.67%, respectively; mean ± SD). A lower concentration of total starch was observed for WSF in comparison to RSF (83.81 and 88.15%, respectively). No differences were observed for damaged starch and ash content between WSF and RSF ($P > 0.05$).

### Flour characterization

Nutritional composition of sorghum flours is reported on dry matter basis in Table 1. A higher content of crude protein (9.95 vs. 8.22%) was observed for WSF compared to RSF, respectively.

**Table 1. Nutrient analysis on dry matter basis of red and white sorghum flour milled at Hall Ross Flour Mill (mean + SD[1]).**

| Property | White Sorghum Flour | Red Sorghum Flour |
|---|---|---|
| Moisture, % | 11.96 ± 0.19 | 12.28 ± 0.47 |
| Dry matter, % | 88.04 ± 0.19 | 87.72 ± 0.47 |
| Crude Protein, % | 9.95[a] ± 0.27 | 8.22[b] ± 0.16 |
| Crude Fat, % | 2.69 ± 0.36 | 3.03 ± 0.23 |
| Total Starch, % | 83.81[b] ± 0.63 | 88.15[a] ± 2.07 |
| Damaged Starch, % | 9.40 ± 1.54 | 8.43 ± 0.66 |
| Total Dietary Fiber, % | 2.47 ± 0.15 | 2.47 ± 0.15 |
| Ash, % | 1.26 ± 0.27 | 1.01 ± 0.20 |
| Sulfur (total), % | 0.09 ± 0 | 0.09 ± 0.055 |
| Phosphorus (total), % | 0.38 ± 0.03 | 0.30 ± 0.02 |
| Potassium (total), % | 0.37[a] ± 0.02 | 0.32[b] ± 0.03 |
| Magnesium (total), % | 0.16 ± 0.006 | 0.13 ± 0.03 |
| Calcium (total), % | 0.017 ± 0.006 | 0.017 ± 0.006 |
| Iron (total), % | 25.20 ± 1.91 | 28.93 ± 4.20 |
| Manganese (total), ppm | 20.17 ± 1.35 | 14.73 ± 3.52 |
| Copper (total), ppm | 2.37[a] ± 0.23 | 1.73[b] ± 0.21 |
| Zinc (total), ppm | 13.20 ± 1.28 | 13.13 ± 2.37 |

[a–b] Means with different superscripts within a row indicate significant difference ($P < 0.05$)

[1] Standard deviation

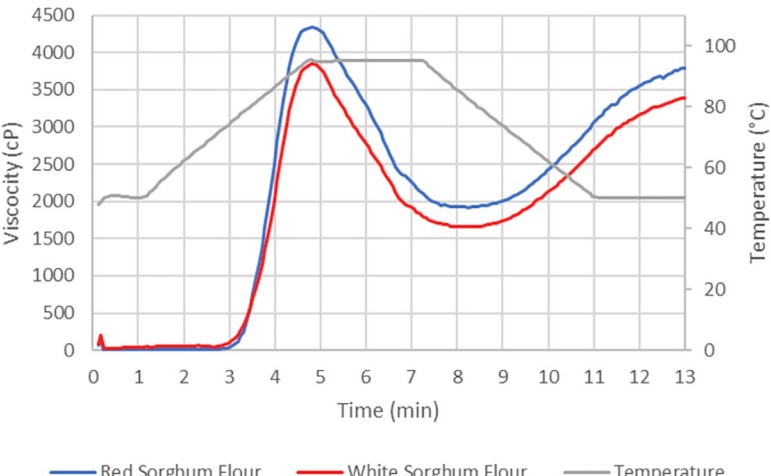

**Fig 2. Pasting profile of white and red sorghum flour used to produce sorghum crisp.**

Total potassium and copper concentration were greater for WSF ($P < 0.05$). No differences were observed for the other nutritional parameters between WSF and RSF ($P > 0.05$).

Although visual differences were noted in the RVA profile graphic (Fig 2), pasting variables were not significantly different between WSF and RSF ($P > 0.05$; Table 2). On the other hand, temperatures of gelatinization were different between WSF and RSF (Table 2). The results revealed a 3.22°C and a 1.47°C increase ($P < 0.05$) in To and Tp, respectively for RSF while Tc and ΔH were not different from WSF.

Particle size parameters of sorghum flours are presented in Table 3. The diameter of WSF varied from 1.09 to 175.05μm and that of RSF from 1.09 to 182.32μm. The d10 and d50 values were greater ($P < 0.05$) for RSF compared to WSF, suggesting that the particle size of RSF is greater than the WSF. No differences were observed for d90 between sorghum flours ($P > 0.05$). Circularity, and convexity did not differ between WSF and RSF ($P > 0.05$). However,

**Table 2. Pasting and thermal properties of white and red sorghum flour (mean ±SD[1]).**

| Property | White Sorghum Flour | Red Sorghum Flour |
|---|---|---|
| Pasting property | | |
| Peak Viscosity, cP | 3769 ± 225 | 4021 ± 181 |
| Trough Viscosity, cP | 1535 ± 78 | 1678 ± 111 |
| Breakdown Viscosity, cP | 2234 ± 196 | 2343 ± 136 |
| Final Viscosity, cP | 3291 ± 120 | 3534 ± 139 |
| Set Back Viscosity, cP | 1570 ± 552 | 1856 ± 61 |
| Peak time, min | 4.76 ± 0.11 | 4.66 ± 0.11 |
| Pasting Temperature,°C | 66.23 ± 11.20 | 69.57 ± 6.25 |
| Thermal property | | |
| Initial temperature, °C | 63.34[a] ± 0.45 | 66.56[b] ± 0.92 |
| Peak temperature, °C | 72.42[a] ± 0.87 | 73.89[b] ± 0.64 |
| Conclusion temperature, °C | 90.86 ± 2.81 | 89.01 ± 1.83 |
| Enthalpy, J/g | 6.49 ± 0.34 | 6.36 ± 0.04 |

[a–b] Means with different superscripts within a row indicate significant difference ($P < 0.0$)

[1] Standard deviation

**Table 3. Morphological parameters of white and red sorghum flour (mean + SD[1]).**

| Parameter | White Sorghum Flour | Red Sorghum Flour |
|---|---|---|
| $d_{10}$, μm | $16.40^b \pm 0.65$ | $17.85^a \pm 0.44$ |
| $d_{50}$, μm | $49.14^b \pm 2.85$ | $62.66^a \pm 4.84$ |
| $d_{90}$, μm | $107.93 \pm 15.82$ | $141.90 \pm 5.20$ |
| Circularity | $0.8470 \pm 0.0085$ | $0.8417 \pm 0.0068$ |
| Aspect ratio | $0.7757^a \pm 0.0075$ | $0.7677^b \pm 0.0087$ |
| Elongation | $0.2243^b \pm 0.0075$ | $0.2323^a \pm 0.0087$ |
| Convexity | $0.9783 \pm 0.0025$ | $0.9777 \pm 0.0006$ |

[a–b] Means with different superscripts within a row indicate significant difference ($P < 0.05$)

[1] Standard deviation

the RSF exhibited a greater ($P < 0.05$) aspect ratio (0. 7757 vs. 0.7677), which is the ratio of particle width to its length, and lower ($P < 0.05$) elongation (0.2243 vs .0.2323) compared to WSF.

## Extrusion process and extrudate characteristics

Bulk density out of the extruder was within target specifications, and similar ($P > 0.05$) for white sorghum crisps (WSC) and red sorghum crisps (RSC; Table 4). Extrusion conditions

**Table 4. Processing data of extruded white and red sorghum flour (mean + SD[1]).**

| Extruder property | White Sorghum Flour | Red Sorghum Flour |
|---|---|---|
| *Pre-conditioner* | | |
| Water, kg/h | $11.43 \pm 0.46$ | $12.13 \pm 0.34$ |
| Steam, kg/h | $59.11 \pm 2.48$ | $59.41 \pm 2.87$ |
| Temperature, ˚C | $48.15 \pm 2.12$ | $47.28 \pm 0.89$ |
| *Extruder* | | |
| Motor load, A | $33.93 \pm 0.47$ | $34.63 \pm 0.81$ |
| Die Temperature, ˚C | $141.48 \pm 21$ | $133.95 \pm 12$ |
| Mass Flow, kg/h | $159.34 \pm 0.64$ | $160.04 \pm 0.62$ |
| *Other data* | | |
| SME[2], kJ/kg | $114.30 \pm 3.33$ | $131.30 \pm 5.67$ |
| IBM[3] (%) | $40.71 \pm 0.72$ | $40.99 \pm 0.82$ |
| Bulk density OE[4], g/L | $108.11 \pm 23.50$ | $95.08 \pm 19.47$ |
| Bulk density OD[5], g/L | $106.70 \pm 21.46$ | $92.16 \pm 23.73$ |
| *Extrudate traits* | | |
| Length, mm | $12.27 \pm 1.25$ | $12.37 \pm 1.62$ |
| Width, mm | $3.98 \pm 1.06$ | $4.08 \pm 1.23$ |
| Weight, g | $0.029 \pm 0.01$ | $0.033 \pm 0.03$ |
| SEI | $6.88 \pm 1.39$ | $7.04 \pm 1.64$ |
| Hardness, N | $12.95 \pm 4.92$ | $14.34 \pm 4.11$ |
| Crispness[6] | $46.62 \pm 21.06$ | $44.38 \pm 17.38$ |

[1] Standard deviation

[2] Specific Mechanical Energy

[3] In-Barrel Moisture

[4] Out of the Dryer

[5] Out of the Extruder

[6] Number of positive peaks.

were kept constant between WSF and RSF (P > 0.05), and no significant differences between the treatments were observed for variables listed. Accordingly, WSC and RSC exhibited similar values for length, width, weigh, and SEI (P > 0.05), and for textural properties (P > 0.05).

## Microbiological testing

Environmental, flour, and extrudate samples were negative for Salmonella as all samples failed to produce typical colonies on selective agars (xylose lysine desoxycholate and bismuth green sulfa).

## Discussion

The white and red sorghum varieties used in this research did not require adjustments during milling to yield similar flour content. The flour yields reported herein are higher than those obtained for red sorghum in laboratory scale by Alvarenga et al. [9] and Moraes et al. [11], demonstrating that commercial milling scale is more efficient than laboratory milling. The flour yield reported in our study was lower compared to the those observed by Alvarenga et al. [9] when red sorghum was milled at commercial scale. The authors milled red sorghum at HRFM and obtained 69.2% yield for flour. Alvarenga et al. [9] produced their flour in one long production test while in our study each sorghum variety was milled in three short-term replicate-days. Shorter milling runs in the current study probably compromised efficiency leading to low flour yield. However, this was necessary to achieve replicates and generate data regarding variation around the process. To our knowledge, this is the first study to report the milling of sorghum in replicates allowing statistical comparison between grain variety.

Characterization of different sorghum flours is important to determine the most suitable application for each ingredient according to its nutritional and physiochemical properties. In the current study, a white sorghum and a red sorghum produced in western Kansas in 2017 were evaluated. However, it is noteworthy that differences in nutritional composition and physiochemical properties between sorghum hybrids are expected; thus, the results observed are applicable for the sorghum hybrids used in this study. The evaluation of WSF and RSF derived from a higher range of grain hybrids produced in Kansas would provide a better characterization of the local grain market. However, it is challenging to produce processing replicates with a high number of treatments. Thus, it was decided to first evaluate one hybrid from each white sorghum and red sorghum to better characterize the milling process.

The nutritional differences between WSF and RSF were in agreement with similar studies that also reported differences in nutritional composition among sorghum hybrids [1, 7, 12]. The range for protein contents (8.22 to 9.95%) are comparable to those found in the literature [1, 7, 12]. The higher protein content measured for WSF was probably due to genetic factors. Total starch, damaged starch, and ash flour content are directly related to the milling process. The total starch results were higher than those of Liu et al. [1] and Winger et al. [7] who reported starch concentration of 78.63% and 72.5%, respectively. The same authors observed a lower damaged starch content (2.7–6.89%) compared those reported in the present study. The severity of the milling process [13], as well as the grain quality, and its preparation can also affect the formation of damaged starch. Harder and larger kernels require more energy input to mill the grain resulting in more deterioration of the starch molecule [14]. Ash is also an important aspect to consider when evaluating flour quality. It is an indication of germ and bran contamination during milling [15]. The ash content observed for WSF and RSF were similar to those of previous studies [1, 7, 12]. The similar content of damaged starch and ash between WSF and RSF suggest that the grain hybrids used in this study would not require different milling conditions in order to produce sorghum flour. Even though ash content was

similar between WSF and RSF, differences in individual mineral contents were observed. The WSF had a higher concentration of total potassium and total copper than RSF. To our knowledge, this is the first study to report and compare mineral content in sorghum flour hybrids. The nutritional differences between WSF and RSF observed in our study are possibly be due to differences in grain composition, genetic and environmental factors. Variability within sorghum hybrids is commonly reported and is probably due to aforementioned factors.

The RVA profile of a sample reflects its physicochemical property [16]. The pasting characteristics observed for WSF and RSF were comparable to those found by Pavalecino et al. [12]. Viscosity parameters are usually a function of starch properties. However, flour samples are composed of other nutrients that can affect their viscosity profile. Although RSF and WSF exhibited different starch and protein content, they resulted in similar pasting properties. The RSF had a higher numerical value for PV and SBV, and PT. Boudries et al. [17] observed higher PV for red sorghum flour compared to white sorghum flour and attributed this difference due to environmental and genetic conditions. Statistical differences were not observed in the current study due to high variably within treatments. This suggests that a larger number of samples may be required to better characterize pasting properties of sorghum flour. Thermal properties of starches can be assessed by DSC and are influenced by a number of factors such as degree of crystallinity, amylose, and amylopectin structure [18]. These parameters might provide a better understanding of thermal properties; however, none of these were assessed in this study. Gelatinization temperatures and enthalpy for WSF and RSF were within the range reported in the literature [7, 17]. The RSF exhibited higher To and Tp compared to WSF which is in agreement with results obtained by Boundries et al. [17]. This may be due to higher degree of crystallinity and higher amylose content for RSF, which hinders water absorption and heat penetration by starch molecules due to a better packing structure. Moreover, the higher PT observed in the pasting profile for both sorghum flour varieties compared to their To are in agreement with Wang et al. [18]. The PT temperature is defined by the initial increase in viscosity due to granule disruption while the To indicates the start of granule swelling. This indicates that starch particles gelatinize before the increase in viscosity [19].

A particle size criterion is used by the Food and Drug Administration to characterize flour, in which "not less than 98% passes through a cloth have openings not larger than 212 um" [20]. In our study, both RSF and WSF met the standard defined by FDA. However, a wide range of particle sizes can be achieved under 212 um which can impact flour quality. Particle size can affect pasting properties of sorghum flour, quality of products such as bread [21], and nutrient digestion rate [22]. In this study, WSF exhibited a lower value for d10 and d50 compared to RSF. Although values for d90 were not statistically different between WSF and RSF, the numerical difference is noteworthy. The lack of statistical difference is probably due to a high variation within WSF; wherein, samples milled on day 1 (124.9 um) had higher d90 compared to day 2 (93.6 um) and day 3 (105.3 um). Although we aimed to achieve a steady state of the milling process before sample collection, the discrepancy in day 1 values for d90 may be a result of machinery adjustments during processing. These results suggest that the WSF presented a lower particle size compared to RSF. In this study, we also evaluated the morphological characteristics of sorghum flour. With an increase in irregularity, the circularity value deviates more from 1. For WSF and RSF the values varied between 0.84–0.86. The higher elongation and lower aspect ratio values in RSF indicate that this sorghum variety presented more elongated and less circular particles (irregularly shaped) compared to WSF. This agrees with the small diameter observed in WSF compared to RSF. Saad et al. [10] also reported in wheat powders that small particles have more regular shape compared to bigger ones. To the best of our knowledge, there are no other published studies reporting the morphology characteristics of sorghum flours. Future studies should further evaluate particle shape of different sorghum

flour varieties to better understand how it can impact flour quality, and thus, final product characteristics.

Extrusion is a high temperature, short time process in which food materials are thermo-mechanically cooked under a combination of temperature, pressure, moisture and mechanical shear. In the current study, extrusion was evaluated as a potential process to produce extruded crisps from WSF and RSF. It was our intention to remove bran during the milling of WS and RS in order to decrease the fiber content and produce well expanded crisps with a smooth texture. Insoluble fibers can rupture the cell walls and prevent air bubbles from expanding [23]. They also have a better affinity to water than starch, restricting water loss at the die and compromising expansion. [24]. The higher starch and lower protein content of RSF compared to WSF did not require different extrusion process parameters. The IBM reported in the current study is higher than those reported in the literature for production of expanded products. Mesquita et al. [25] evaluated the effect of moisture content on extrusion of sour cassava starch and flaxseed flour blends up to 20% moisture, and they achieved a good product when moisture was added at 12%. A similar feed moisture content (16–21%) was reported by Baik et al [26] when evaluating extrusion of barley flour for production of expanded cereals. However, the previous authors did not have a pre-conditioner in advance of the extrusion process. The high steam addition into the pre-conditioner resulted in high IBM in our study. This was required to hydrate the material and improve its fluidity as the small particle size of flour can impair flow and cause clumping. Steam injection rates were recorded from the control system. However, due to the lower rates, both the preconditioner and extruder steam injection zones were operated via the condensate relief valves which were open continuously in an effort to reduce additional moisture from entering the process. With this operational constraint, it was necessary to reduce the steam injection rates by approximately 1% to 2% from the recorded values in future trials. Temperature in the preconditioner was lower than expected, and we believe this may have been due to higher steam loss in the system and (or) inaccurate readings from the temperature probe attached to the preconditioner discharge. The lubricating effect of water can decrease shear, and thereby reduce SME input and product expansion. However, both WSC and RSC expanded well, and had a high SEI. This may be due to the high temperature at the end of the barrel, high screw speed, and aggressive screw configuration. Previous authors [26] reported a lower SEI for production of an expanded cereal using barley flour. However, the different experimental setups and equipment used make it difficult to directly compare results among studies. Although not statistically different, extrusion of RSF exhibited higher SME and led to a less dense product. This variance may be a result of inconsistent feed rate into the system. This could have been assessed by measuring material flow rate out of the extruder rather than calculating it. Steam losses and inconsistent delivery of material were not considered when mass flow rate was assessed in this study. This could potentially overestimate actual mass flow rate. A lower feed rate for RSF into the system may have resulted in the higher numerical SME, even though it was not statistically different from WSF.

Hardness and crispness are important texture measurements to be assessed in expanded products. They are related to expansion and cell structure development within the starch matrix during extrusion [27]. In the current study, hardness was defined as the peak force required to disintegrate the extrudate, and crispness was defined as the number of positive peaks during deformation of the material by the probe. Hardness of the product has been positively correlated with addition of water into the system; an increase in moisture content may reduce expansion, and thus increase product hardness [28]. On the other hand, increasing feed moisture content was associated with decreased crispness of rice extrudates as it can compromise bubble growth and result in a less expanded product [29]. A product with a lower degree of bubble growth has bigger cells with ticker walls which can decrease product

crispness. As both WSC and RSC were produced under the same processing conditions, no differences were observed between texture properties. Duizer et al. [30] evaluated hardness and crispness of corn-based crisps using a trained sensory panel, and with a bite force apparatus. In their study, a negative relationship was observed between the maximum bite force and crispness of extrudates. The authors also found a higher value for hardness and crispness compared to our results. This could be due to the different methodologies used between studies. The use of trained panelists to evaluate product characteristics could have provided more detailed information regarding sensory and textural aspects of white and red sorghum crisps. For example, detailed properties of crispness such as sound production during crushing the product could have been assessed. Future studies should consider the use of trained panelists in addition to evaluation with a texture analyzer to yield a better understanding of sensory properties.

Microbiological assessment of flour mill and end products–flour and crisp–was performed to assure food safety. An outbreak of Salmonella Typhimurin Phage Type 42b was reported in New Zealand due to contamination of raw wheat flour with these bacteria [31]. Thus, it is important to monitor microbial contamination during processing. In the current study, environmental and food samples were all negative for Salmonella indicating that the production chain of sorghum crisp was completely free from this group of bacteria. It is noteworthy that extrusion is considered a kill step due to its high temperature, but extruded product may be contaminated post extrusion during the drying and packing. Purchasing grains from trustworthy supplies, and monitoring processing conditions are examples of practices to prevent contamination in the process.

## Conclusions

Characterization of milling process of different sorghum hybrids, as well as flour quality are essential to establish the product in the market. In the current study, the milling of red and white sorghum yielded similar flour content, but differences were observed between chemical composition and thermal properties of respective flours. White and red sorghum flour required similar extrusion conditions in order to produce expanded sorghum crisp, thereby extrudates exhibited similar macrostructure and textural characteristics. Sorghum crisps derived from extrusion of white and red sorghum flour are promising products to boost the use of sorghum in the market.

## Supporting information

**S1 File.**
(XLSX)

## Author Contributions

**Conceptualization:** Charles Gregory Aldrich.

**Data curation:** Julia Guazzelli Pezzali, Anu Suprabha-Raj, Kaliramesh Siliveru.

**Formal analysis:** Julia Guazzelli Pezzali, Anu Suprabha-Raj, Kaliramesh Siliveru.

**Funding acquisition:** Charles Gregory Aldrich.

**Investigation:** Charles Gregory Aldrich.

**Methodology:** Charles Gregory Aldrich.

**Project administration:** Julia Guazzelli Pezzali, Charles Gregory Aldrich.

**Resources:** Charles Gregory Aldrich.

**Software:** Julia Guazzelli Pezzali, Anu Suprabha-Raj.

**Supervision:** Charles Gregory Aldrich.

**Validation:** Julia Guazzelli Pezzali, Charles Gregory Aldrich.

**Visualization:** Julia Guazzelli Pezzali.

**Writing – original draft:** Julia Guazzelli Pezzali.

**Writing – review & editing:** Anu Suprabha-Raj, Kaliramesh Siliveru, Charles Gregory Aldrich.

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
