## [Decision Letter · Decision Letter 0]

29 Apr 2020

PONE-D-20-08606

Characterization of white and red sorghum flour and their potential use for production of extrudate crisps

PLOS ONE

Dear Dr. Aldrich,

Thank you for submitting your manuscript to PLOS ONE. After careful consideration, we feel that it has merit but does not fully meet PLOS ONE’s publication criteria as it currently stands. Therefore, we invite you to submit a revised version of the manuscript that addresses the points raised during the review process.

We would appreciate receiving your revised manuscript by Jun 13 2020 11:59PM. To enhance the reproducibility of your results, we recommend that if applicable you deposit your laboratory protocols in protocols.io, where a protocol can be assigned its own identifier (DOI) such that it can be cited independently in the future. For instructions see: http://journals.plos.org/plosone/s/submission-guidelines#loc-laboratory-protocols

We look forward to receiving your revised manuscript.

Kind regards,

Leonidas Matsakas

Academic Editor

PLOS ONE

Journal Requirements:

Reviewers' comments:

Reviewer's Responses to Questions

**Comments to the Author**

1. Is the manuscript technically sound, and do the data support the conclusions?

Reviewer #1: Yes

Reviewer #2: Yes

2. Has the statistical analysis been performed appropriately and rigorously? 

Reviewer #1: Yes

Reviewer #2: Yes

3. Have the authors made all data underlying the findings in their manuscript fully available?

Reviewer #1: Yes

Reviewer #2: Yes

4. Is the manuscript presented in an intelligible fashion and written in standard English?

Reviewer #1: Yes

Reviewer #2: Yes

5. Review Comments to the Author

Reviewer #1: This manuscript is well written and technically sound. Some questions and comments to be addressed before publication follow below.

This study relies on only two samples (which the authors have acknowledged in the discussion) and this should be addressed in the abstracted and early in the manuscript. Suggest adding "one white and red..." on line 6 and "The white and red sorghum..." on line 7. On line 10, revise to read "The white and red sorghum used in this study..." At line 210 revise to "The white and red sorghum varieties used in this research..." At line 214 change to "...Alvarenga et al [9] when a red sorghum.." On line 223 change to "...current study, a white sorghum and a red sorghum..."

Lines 236-239. Related to starch damage and grain hardness, it would be beneficial to include grain hardness measurements of the two samples used in this study to relate hardness to starch damage. This could also be inferred from the particle size data as particle size has been used to measure grain hardness.

Line 246- and other places in the publication. The authors have stated that nutritional differences may exist and have hinted this may be related to grain color. However, pericarp color is under control of specific genes and has never been linked to endosperm composition (see chapter of W. Rooney in Handbook of Sorghum Technology which lists known genes and their impacts or the work of Lloyd Rooney and pericarp color and genetic controls). There is no evidence to support the idea that overall grain composition/nutritional profile is may be linked to or influenced grain color (other than phenolic content and composition), thus the manuscript needs to be revised to remove those suggestions.

Lines 252-254. There are many, many book chapters that review sorghum grain composition and all show the variability that sorghum can have in grain composition (protein, starch, fat, phenolics, etc...). Thus it is misleading to state that the literature contains inconsistent results...the variability present in the literature is due to the variability within sorghum which is very well documented at the genetic and chemical level. This statement needs to be revised to more clearly state the variability that can be present in sorghum.

Lines 269-270. Consider measuring amylose levels and add to the manuscript to support this speculation.

Reviewer #2: This study analyze the milling process and flour quality of white and red sorghum and their extrusion as a potential process to produce sorghum crisps.

I consider this idea is very valuable. However, the paper needs improvements, some important information is missing and some results only a limited explanation is given.

The objective to measured salmonella should be described/discussed in introduction section. I wonder if the salmonella analyzes and results are a necessary part of this work?

The objective of this study was to evaluate the milling process and flour quality of white and red sorghum and evaluate extrusion as a potential process to produce sorghum crisps.

However, in the opinion of this reviewer no variable conditions of milling process were tested to gain knowledge. Tempering and milling conditions were chosen but clear explanation should be done about the election of processing variables. Why 16.5 % and 19 h were used? Are there these condition optimized?

Since the objective of this study was to evaluate the milling process, did the authors tested different conditions? Clarify

L64. Milling process stabilization?

Did the authors make more than one testing/analysis on flour sample evaluations? Or only one test of each lot? Clarify

L82. It is not clear what (7+2 mg) means. Clarify

Extrusion process.

Include barrel diameter and barrel length to diameter ratio

Was the extruder barrels heat by any mean? Clarify

Did the extruder have temperature probes in the barrel? Clarify

All the steam was added in the pre-conditioner?

Where the temperature of pre-conditioner was measured? In the opinion of this reviewer the temperature (47-48°C) of the pre-conditioner is too low according to the amount of steam addition

How was the extruder configuration chosen? Did the authors try different configurations?

Change hr by h (hours)

L135-138. The order of the statement should be change as follow:

A compression test was performed using a 25 mm cylindrical probe at a pre-test speed of 2 mm/s, test speed of 2 mm/s, a post-test speed of 10 mm/s, and strain level of 90%. The first peak fracture force, and the number of positive peaks were taken as a measure of hardness and crispness, respectively.

Each extrudate was measured alone? Clarify

Only flour from HRFM was tested for salmonella? Extrudates?

Results

L157-159. It is not clear for this reviewer what do the authors mean with:

“13.74 + 0.41 to 15.90 + 0.99 % (wb), and from 13.38 + 0.31 to 15.21 + 0.73 % (wb) for WS and RS, respectively”

“(59.03 + 8.676 and 57.02 + 2.67 %, respectively”. Clarify

L224-225. “However, it is noteworthy that hybrids within each sorghum variety may have different nutritional and physiochemical properties.”

This phrase is not clear for this reviewer. Do the authors know the hybrids and/or varieties evaluated?

Did the author evaluate the flour/extrudates color parameters? Were there any differences among samples?

L307-315. There are some points that needs more discussion: high steam but low temperature in the pre-conditioner, very high IBM. Please discuss more deeply.

L318. Please discuss and compare the screw configuration used. Why is aggressive? How aggressive is?

L332-334. Could the authors explain these behaviors? Or expand the discussion of the cited references

As the authors have mentioned in the discussion, the lack of statistical differences were not observed in the current study due to high variably within treatments, which in turn depend of experiment design. In the opinion of this reviewer it should be mentioned in conclusions also

6. PLOS authors have the option to publish the peer review history of their article (what does this mean?). If published, this will include your full peer review and any attached files.

Reviewer #1: No

Reviewer #2: No

---

## [Author Response · Author response to Decision Letter 0]

25 May 2020

Reviewer #1: This manuscript is well written and technically sound. Some questions and comments to be addressed before publication follow below.

This study relies on only two samples (which the authors have acknowledged in the discussion) and this should be addressed in the abstracted and early in the manuscript. Suggest adding "one white and red..." on line 6 and "The white and red sorghum..." on line 7. On line 10, revise to read "The white and red sorghum used in this study..." At line 210 revise to "The white and red sorghum varieties used in this research..." At line 214 change to "...Alvarenga et al [9] when a red sorghum.." On line 223 change to "...current study, a white sorghum and a red sorghum..."

As suggested by the reviewer, minor changes (which are noted in track-changes) throughout (abstract, introduction, discussion) the manuscript were made to address the single variety of each white and red sorghum used in this study. 

Lines 236-239. Related to starch damage and grain hardness, it would be beneficial to include grain hardness measurements of the two samples used in this study to relate hardness to starch damage. This could also be inferred from the particle size data as particle size has been used to measure grain hardness.

We agree that measuring grain hardness might further corroborate the inferences made in the current study related to starch damage and particle size. However, since this relationship had already been established by other researchers, wherein harder kernels require more energy to break, and thus result in more severe milling process. We would like to focus on more direct measures and to focus on the outcomes in the final product: e.g., starch damage and particle size. Which is the goal of this work. 

Line 246- and other places in the publication. The authors have stated that nutritional differences may exist and have hinted this may be related to grain color. However, pericarp color is under control of specific genes and has never been linked to endosperm composition (see chapter of W. Rooney in Handbook of Sorghum Technology which lists known genes and their impacts or the work of Lloyd Rooney and pericarp color and genetic controls). There is no evidence to support the idea that overall grain composition/nutritional profile is may be linked to or influenced grain color (other than phenolic content and composition), thus the manuscript needs to be revised to remove those suggestions.

We respectfully concede that the gross nutritional composition is not a direct effect of coat color. Therefore, the manuscript was revised to remove suggestions implying that grain color could be leading to differences in nutritional composition. 

Lines 252-254. There are many, many book chapters that review sorghum grain composition and all show the variability that sorghum can have in grain composition (protein, starch, fat, phenolics, etc...). Thus it is misleading to state that the literature contains inconsistent results...the variability present in the literature is due to the variability within sorghum which is very well documented at the genetic and chemical level. This statement needs to be revised to more clearly state the variability that can be present in sorghum.

This statement was revised, and we rephrased the sentence in which we mentioned inconsistent results in the literature. 

Lines 269-270. Consider measuring amylose levels and add to the manuscript to support this speculation.

Like grain hardness and starch damage above, we agree that measuring amylose and amylopectin levels would provide additional details to the manuscript. Although we felt that this was “nice-to-have” information and was not critical to the interpretation of the results. Though, we did address this limitation in the paper with the suggestion that future studies perform this analysis. Especially if one were to evaluate a broader cross section of varieties, crop years, or growing regions. Further, the impact on gelatinization was minimal and the enthalpy was similar between the red and white sorghum flour. 

Reviewer #2: This study analyze the milling process and flour quality of white and red sorghum and their extrusion as a potential process to produce sorghum crisps.

I consider this idea is very valuable. However, the paper needs improvements, some important information is missing and some results only a limited explanation is given.

The objective to measured salmonella should be described/discussed in introduction section. I wonder if the salmonella analyzes and results are a necessary part of this work?

We have added a sentence regarding Salmonella evaluation in the final product. Since a major recall in all-purpose flour was issued last year due to possible Salmonella contamination, we believe it was valuable data to report. By including this data, we also report the methodology used which can be applied in an industry setting, as well as future studies. Because there are not many published manuscripts reporting the approach (sample sites, etc) for evaluating Salmonella contamination in the milling process, we believe it is valuable to include this data. Furthermore, it does not influence the other results. Finally, these crisps were to be included in food products for animal and human sensory evaluation and therefore it was our desire to assure that the material was safe. 

The objective of this study was to evaluate the milling process and flour quality of white and red sorghum and evaluate extrusion as a potential process to produce sorghum crisps.

However, in the opinion of this reviewer no variable conditions of milling process were tested to gain knowledge. Tempering and milling conditions were chosen but clear explanation should be done about the election of processing variables. Why 16.5 % and 19 h were used? Are there these condition optimized?

Since the objective of this study was to evaluate the milling process, did the authors tested different conditions? Clarify

The moisture target and conditioning time were based on previous sorghum runs in the same facility. Alvarenga et al (2018) tempered red sorghum grain in the same unit for 24h to achieve a moisture content of 16%. We used her dataset as a starting point, and the milling operator was confident we would be able to temper the sorghum grain for 19h to achieve a moisture content of around 16%. Although previous sorghum production-runs were performed at this milling facility, our study was the first-time replication with multiple days occurred. We have added a sentence in the methodology section stating that the tempering conditions were based on previous production-runs to provide additional clarity. We also reported the number of break and sizing passages and the % release, and we mentioned the use of those settings from previous research. The intent was to be consistent with the parameters reported by other researchers in order to characterize the milling process utilized. 

L64. Milling process stabilization?

After tempering, each sorghum was milled for 30 minutes before sample collection – which was called was process stabilization, aka steady-state. It takes some amount of time to get the process to run smoothly and to avoid product fluctuations. Thus, we waited 30 minutes (milling process stabilization) before starting sample collection. A sentence was add (New L 63) to describe this.

Did the authors make more than one testing/analysis on flour sample evaluations? Or only one test of each lot? Clarify

This was described on new L68-70: “Upon process stabilization within each replicate production cycle, flour samples were collected every 30 minutes totaling three subsamples per day (replicate) for each sorghum variety. Subsamples within day were composited and prepared for analysis as described below.” 

L82. It is not clear what (7+2 mg) means. Clarify

We changed the symbol (7 ± 2mg) to aid in clarification. A sample weight between 5 to 9 mg was added into the aluminium pan. 

Extrusion process.

Include barrel diameter and barrel length to diameter ratio

New L100 - Information has been included 

Was the extruder barrels heat by any mean? Clarify

New L106 - All extruder barrel sections (jacketed) were heated with steam with the exception of the last section (head) which was cooled to prevent more expansion of the product. This information was included. 

Did the extruder have temperature probes in the barrel? Clarify

The temperature probe was added at the extrudate exit in the die. We added this information in the methodology to ad clarity. 

All the steam was added in the pre-conditioner?

Steam was only added in the preconditioner. We included a sentence specifying that there was no steam addition in the extruder barrel.

. 

Where the temperature of pre-conditioner was measured? In the opinion of this reviewer the temperature (47-48°C) of the pre-conditioner is too low according to the amount of steam addition

New L 333 - The temperature was measured at the discharge. The temperature was recorded from the control system. Unfortunately, we did not manually measure the discharge temperature of the product to ensure accuracy. We do agree that the temperature was not as high as expected and have included possible explanations in the discussion regarding this point 

How was the extruder configuration chosen? Did the authors try different configurations?

Extruder screw profile was based on preliminary trial on the same single-screw extrusion process system. We added this information to aid in clarification. 

Change hr by h (hours)

Hr was replaced by h. Thank you for picking up this detail.

L135-138. The order of the statement should be change as follow:

A compression test was performed using a 25 mm cylindrical probe at a pre-test speed of 2 mm/s, test speed of 2 mm/s, a post-test speed of 10 mm/s, and strain level of 90%. The first peak fracture force, and the number of positive peaks were taken as a measure of hardness and crispness, respectively.

Order was changed as suggested by the reviewer. 

Each extrudate was measured alone? Clarify

Twenty one extrudates were individually assessed from each replicate. This sentence has been updated to provide better clarity. 

Only flour from HRFM was tested for salmonella? Extrudates?

Flour AND extrudate samples from each sorghum variety were tested for Salmonella (L152).

Results

L157-159. It is not clear for this reviewer what do the authors mean with:

“13.74 + 0.41 to 15.90 + 0.99 % (wb), and from 13.38 + 0.31 to 15.21 + 0.73 % (wb) for WS and RS, respectively”

“(59.03 + 8.676 and 57.02 + 2.67 %, respectively”. Clarify

They represent the mean +/- standard deviation. The has been added to the text.

L224-225. “However, it is noteworthy that hybrids within each sorghum variety may have different nutritional and physiochemical properties.”

This phrase is not clear for this reviewer. Do the authors know the hybrids and/or varieties evaluated?

The sentence was modified to aid in clarity. We just wanted to clarify to the reader that the results observed are valid for the hybrid used in the study because variation between hybrids is expected. We reported the location and crop year for both the white and red sorghum used in the trial and provided discussion that the use of more than one white or red sorghum variety might be beneficial. However, it is difficult to source a high number of grain varieties and test it in large production scale projects as was done in this study.

Did the author evaluate the flour/extrudates color parameters? Were there any differences among samples?

We did not evaluate the color of the final product in this study as it was not our intention to assess consumer acceptance. This was assessed in the crisps in a study conducted afterward (Pezzali et al., 2020 in progress). In that work we analyzed the use of the white and red sorghum crisps produced in this study in a granola bar application and we hope to have that work published in a separate manuscript in the near future. That paper will provide much more detail regarding visual characteristics of the product and consumer acceptance. Due to the size and scope of the work it was not possible to combine both manuscripts and would distract from the processing focus of this manuscript.

L307-315. There are some points that needs more discussion: high steam but low temperature in the pre-conditioner, very high IBM. Please discuss more deeply.

New L329, New L347 - More detail was included in the discussion regarding these process parameters. 

L318. Please discuss and compare the screw configuration used. Why is aggressive? How aggressive is?

The screw configuration was determined from previous proof of concept research and a variety of trial and error efforts to produce the crisps. Degree of “aggressive” tends to be more slang and relative to the individual machine selected. We did provide information on the screw configuration and all the parameters used during production. We based the discussion on the common approach used in published papers reporting extrusion processing (i.e., Zhu et al., 2010; Devi et al., 2013; Razzaq e tal., 2012; Pezzali and Aldrich, 2019). It is challenging to compare screw configurations among different studies because of the variety of extruders used (single vs. twin screw), different extruder sizes (pilot scale vs. commercial scale; and with many differences within each) which impact the magnitude of shear imparted to the product. We hope that reporting the screw profile provides sufficient guidance for future studies and interpretation of our results. 

L332-334. Could the authors explain these behaviors? Or expand the discussion of the cited references

New L 354 - More explanation was provided as suggested by the reviewer. 

As the authors have mentioned in the discussion, the lack of statistical differences were not observed in the current study due to high variably within treatments, which in turn depend of experiment design. In the opinion of this reviewer it should be mentioned in conclusions also

This is the first study in which the milling of sorghum was done in replicates in a large-scale production facility. A complete replication requires shut down of the equipment. If the equipment was not shut down between collections it would not be a true replicate, but rather repeated measures. It is extremely difficult to achieve replications in a large-scale production due to cost and high volume of raw material required. Thus, although a higher number of samples may be desired to decrease variability and to demonstrate statistical differences, the fact that we were able to achieve three replications in both milling and extrusion is extremely valuable and representative of research in this field. Thus, we do not think that other statistical approaches would increase the confidence surrounding the outcome. We have addressed this in the discussion to wit a larger sample size may be beneficial. However, that we achieved replication within the challenges of full-scale production facilities/equipment was significant in and of itself.

---

## [Decision Letter · Decision Letter 1]

5 Jun 2020

Characterization of white and red sorghum flour and their potential use for production of extrudate crisps

PONE-D-20-08606R1

Dear Dr. Aldrich,

We’re pleased to inform you that your manuscript has been judged scientifically suitable for publication and will be formally accepted for publication once it meets all outstanding technical requirements.

Kind regards,

Leonidas Matsakas

Academic Editor

PLOS ONE

Additional Editor Comments (optional):

Reviewers' comments:

Reviewer's Responses to Questions

**Comments to the Author**

1. If the authors have adequately addressed your comments raised in a previous round of review and you feel that this manuscript is now acceptable for publication, you may indicate that here to bypass the “Comments to the Author” section, enter your conflict of interest statement in the “Confidential to Editor” section, and submit your "Accept" recommendation.

Reviewer #1: All comments have been addressed

Reviewer #2: All comments have been addressed

2. Is the manuscript technically sound, and do the data support the conclusions?

Reviewer #1: Yes

Reviewer #2: Yes

3. Has the statistical analysis been performed appropriately and rigorously? 

Reviewer #1: Yes

Reviewer #2: Yes

4. Have the authors made all data underlying the findings in their manuscript fully available?

Reviewer #1: Yes

Reviewer #2: Yes

5. Is the manuscript presented in an intelligible fashion and written in standard English?

Reviewer #1: Yes

Reviewer #2: Yes

6. Review Comments to the Author

Reviewer #1: Comments to my suggestions have been addressed in the revision of the manuscript and the manuscript from my perspective is ready for publication.

Reviewer #2: After reviewing the manuscript for a second time, I note that the manuscript has improved significantly in order to be published.

7. PLOS authors have the option to publish the peer review history of their article (what does this mean?). If published, this will include your full peer review and any attached files.

Reviewer #1: No

Reviewer #2: No

---

## [Editor Report · Acceptance letter]

12 Jun 2020

PONE-D-20-08606R1 

Characterization of white and red sorghum flour and their potential use for production of extrudate crisps 

Dear Dr. Aldrich:

I'm pleased to inform you that your manuscript has been deemed suitable for publication in PLOS ONE. Congratulations! Your manuscript is now with our production department. 

Kind regards, 

on behalf of

Dr. Leonidas Matsakas 

Academic Editor

PLOS ONE